# Use of three points to determine the accuracy of guided implantation

**Ye Liang[1‡], ShanShan Yuan[2‡], JingJing Huan[1], HuiXin Wang[1], YiYi Zhang[1], ChangYun Fang[1]☯*, Jia-Da Li[3]☯***

1 Department of Stomatology, Xiangya Hospital of Central South University, Changsha, Hunan Province, China, 2 Affiliated Stomatology Hospital of Guangzhou Medical University, Guangzhou, Guangdong, China, 3 School of Life Sciences, Central South University, Changsha, Hunan Province, China

☯ These authors contributed equally to this work.
‡ These authors are co-first authors on this work.
* fcy@kuye.cn (CYF); lijiada@sklmg.edu.cn (JDL)

**Data Availability Statement:** All relevant data are within the manuscript and its Supporting Information files.

**Funding:** This work was financially supported by grants from the Hunan Province Science and

## Abstract

### Aim

This study aims to establish an open-source algorithm using Python to analyze the accuracy of guided implantation, which simplifies interstudy comparisons.

### Methods

Given ≥3 landmark pairs, this Tri-Point (TriP) method can register images. With ≥4 landmark pairs, TriP can calculate system errors for image registration. We selected 8 indicators from the literature. Considering development errors in new bone on cone beam computed tomography (CBCT), we added the indicators of apical rectified deviation (ARD) and coronal rectified deviation (CRD), providing accurate references but neglecting depth deviations. Our program can calculate and output these indicators. To evaluate the TriP method's feasibility, an implantation group assisted by a Visual Direction-INdicating Guide (VDING) was analyzed. Accuracy was measured with the traditional and proposed TriP methods. Factors affecting the system error of the method were then analyzed.

### Results

Comparisons with paired-sample t-tests showed that our TriP method was similar to the traditional method in evaluating implantation accuracy, with no significant difference (P>0.05). The average system error was 0.30±0.10 mm when the TriP method evaluated the VDING template. The results showed that increasing the provided landmarks from 4 to 5 pairs decreased the between-group differences significantly (P<0.05). With ≥6 pairs of landmarks, the system error tended to be stable, and the groups showed no statistically significant differences (P>0.05). Large distances between landmarks are helpful in reducing system error, as demonstrated with a geometric method.

Technology Department (Hunan Natural Science Fund - Youth Foundation Project, 2018JJ3850; Key R&D Program of Hunan Province, 2017SK2161), National Natural Science Foundation of China (81901065), Central South University (2018zzts916), and Hunan Health Commission (B2019192). The funders had no role in study design, data collection and analysis, decision to publish, or preparation of the manuscript.

**Competing interests:** The authors have declared that no competing interests exist.

## Conclusions

This study established an open-source algorithm to analyze the accuracy of guided implantation with system errors reported.

## Introduction

As a bridge between virtual and real systems, digital technology-based guide templates can make implantation more precise than manual implantation. Nevertheless, to thoroughly analyze the template or further increase the accuracy, we need to further analyze the accuracy of guided implantation.

At present, some commercial software programs provide methods for analyzing accuracy with the original data in DICOM or STL files [1,2,3]. However, most of the algorithms are not public and provide no system error. Additionally, because most commercial software is expensive, most studies have used only one software package to calculate the accuracy [1,4], making it difficult to evaluate the measurement accuracy of the analysis itself.

Additionally, analyses of implantation accuracy must include calculations of related indicators. Some commonly used indicators include coronal global deviation (CGD), apical global deviation (AGD), and angular deviation (AD). However, implantation descriptions involving just these 3 indicators have shortcomings. Therefore, some studies have added depth and horizontal deviation [5]. Furthermore, some studies have used nontraditional indicators [6,7]. Moreover, few indicators describe the implanted cover screw and healing abutment, which are ignored. Therefore, better indicators are needed.

Differences between methods and indicators render it difficult to compare different research results. To solve this problem, we attempt to establish an open-source method for analyzing implantation accuracy that can compute system errors. Simultaneously, we screened some relatively mature evaluation indicators of implant accuracy by referring to previous studies and added some indicators based on our research. We hope that our open-source software will enable direct calculation of the results of these indicators.

In this study, we applied this open-source software to a guide template called Visual Direction-INdicating Guide (VDING) and compared our software with the traditional commercial software Mimics. The guide template was modified to address the lack of angular restriction provided by the short guide sleeves used when the occlusal distances at implant sites are slightly insufficient for normal height templates. Because the VDING template adds a visual direction-indicating device, the operator can more easily and objectively observe the angle of the drill needle in real time without relying entirely on the mechanical structure of the guide sleeve.

We hope that our open-source program can better promote digital technology and help further improve the accuracy of implant surgery.

## Materials and methods

### Ethics statement

The study itself excluded the involvement of human beings, nor did it cause harm to human beings. However, clinical examination data were partly used in the study, and the use of these data was approved by Ethics Commitee of Xiangya Hospital of Central South University, approval number: 201512515. All Patients provided written informed consent to the use of CBCT data in this research.

## Tri-Point (TriP) image registration method

The data for the planned and real implants are in preoperative and postoperative cone bean computed tomography (CBCT) images, respectively, so the positions of the planned and real implants cannot be directly compared.

To compare the differences, we need to align the two images via features in the images to geometrically transform the two image spaces. Our Tri-Point (TriP) method is established for researchers who TRY to test the IMPLANT PLAN accuracy. This method defines a new coordinate system by using relatively unchanged anatomical sites before and after surgery, allowing comparison of the positions of the planned and real implants.

## Space vector-based relative coordinate system

The coordinate system $\Sigma$ has 3 noncollinear points with coordinates $A(X_1,Y_1,Z_1)$, $B(X_2,Y_2,Z_2)$, and $C(X_3,Y_3,Z_3)$ (Fig 1A).

To establish a relative coordinate system, we define A as the origin and the vector $\overrightarrow{AB}$ as the positive direction of the X-axis. Therefore, the X-axis unit vector $\overrightarrow{x}$ can be obtained as:

$$\overrightarrow{x} = \frac{\overrightarrow{AB}}{|\overrightarrow{AB}|} \qquad (2-1)$$

The noncollinear points A, B, and C can define a plane (Fig 1B). Points $C_x$ and $C_y$ are the projections of point C on the X-axis and the Y-axis, respectively. $\theta$ is the value of $\angle CAB$.

We set the Y-axis in the ABC plane, so point C exists in the XAY plane and satisfies:

$$\overrightarrow{AC_y} = \overrightarrow{AC} - |\overrightarrow{AC}| \cdot \frac{\overrightarrow{AB} \cdot \overrightarrow{AC}}{|\overrightarrow{AB}| \cdot |\overrightarrow{AC}|} \qquad (2-2)$$

Therefore, we can obtain the unit vector $\overrightarrow{y}$ of the Y-axis as:

$$\overrightarrow{y} = \frac{\overrightarrow{AC_y}}{|\overrightarrow{AC_y}|} \qquad (2-3)$$

We set $\overrightarrow{n}$ as the normal vector of plane ABC, so this vector satisfies:

$$\overrightarrow{n} = \overrightarrow{AB} \times \overrightarrow{AC} \qquad (2-4)$$

The unit vector of the Z-axis is obtained as:

$$\overrightarrow{z} = \frac{\overrightarrow{n}}{|\overrightarrow{n}|} \qquad (2-5)$$

The obtained unit vectors, $\overrightarrow{x}$, $\overrightarrow{y}$, $\overrightarrow{z}$, of the three coordinate axes can be used to establish a three-dimensional rectangular coordinate system, $\Phi$ (Fig 1C).

## Conversion from the original coordinate system to a relative coordinate system

Point G is the point targeted for conversion, and its coordinates are $G(X_4,Y_4,Z_4)$ in coordinate system $\Sigma$ and $G'(X_4',Y_4',Z_4')$ in coordinate system $\Phi$. The spatial relationship between point G' and points ABC remains the same as the relationship for point G (Fig 1D and 1E).

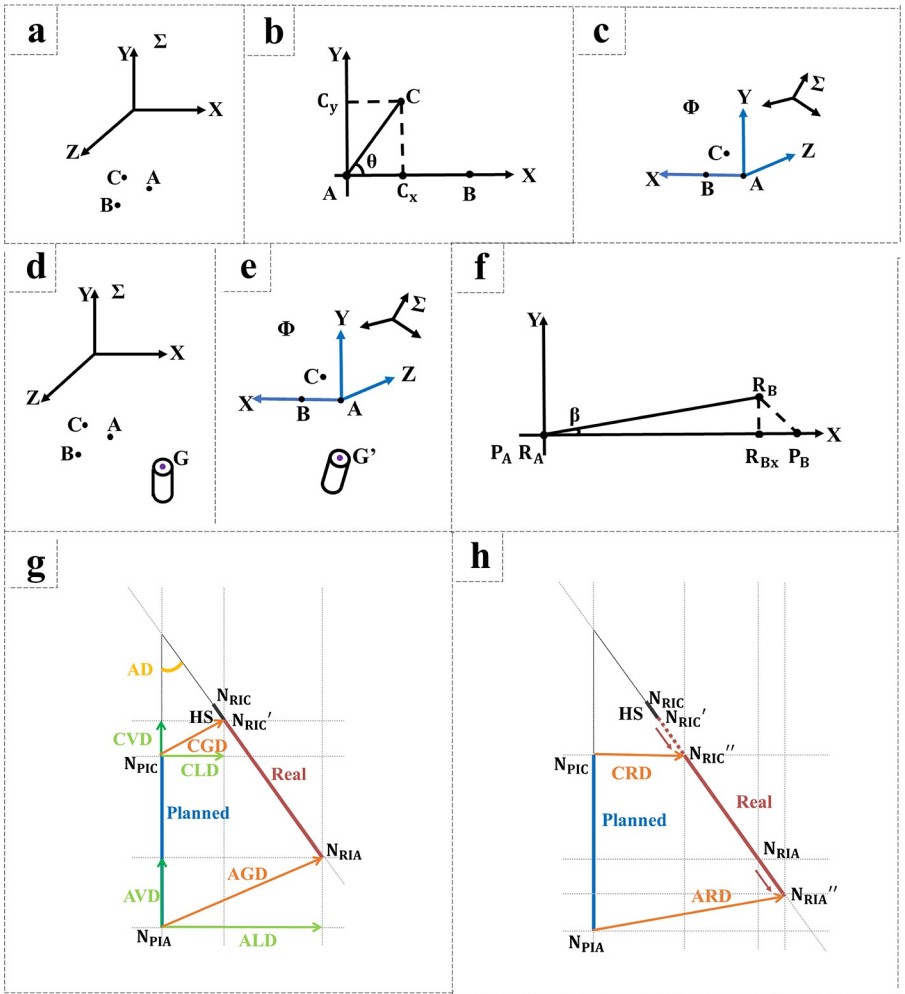

**Fig 1. Illustrations of geometric relationships.** (A) Three-dimensional rectangular coordinate system Σ and three points, A, B, and C, in the system. (B) A plane (plane XAY) determined by the points A, B, and C. (C) The coordinate system Φ determined by the points A, B, and C. (D) The basic points A, B, and C and the target point G in coordinate system Σ. (E) G' point in coordinate system Φ after geometric transformation. (F) Angular deviation in registration. (G) Indicators to evaluate implantation accuracy. (H) Indicators after rectification of coronal depth.

G'$(X_4', Y_4', Z_4')$ is solved with Eqs (2-1) to (2-5). These equations indicate that point G' satisfies:

$$\overrightarrow{AG} = X_4' \cdot \overrightarrow{x} + Y_4' \cdot \overrightarrow{y} + Z_4' \cdot \overrightarrow{z} \qquad (2-6)$$

The coordinates of 4 points in coordinate system Σ are A$(X_1,Y_1,Z_1)$, B$(X_2,Y_2,Z_2)$, C$(X_3,Y_3, Z_3)$, and G$(X_4,Y_4,Z_4)$. The 3 coordinate axes of the coordinate system obtained above are $\overrightarrow{x} = (X_X, X_Y, X_Z)$, $\overrightarrow{y} = (Y_X, Y_Y, Y_Z)$, and $\overrightarrow{z} = (Z_X, Z_Y, Z_Z)$. By substituting these known values into Eq (2-6), the value of $X_4'$, $Y_4'$ and $Z_4'$ are obtained. The solution of G'$(X_4',Y_4',Z_4')$ is the coordinates of the original point G in coordinate system Φ.

The points in the planned image are in coordinate system P, and the points in the real implant image are in coordinate system R. In each coordinate system, the corresponding anatomical landmarks L1, L2, and L3 are selected as the basic points A, B and C described above. These three pairs of basic points are anatomically congruent. Thus, coordinate systems P and

R are converted to the same coordinate system, N. The detailed process is described in Appendix 1.

## Error evaluation and option of landmark points

Errors occur when selecting each landmark point in images. Thus, using different landmarks as basic points in steps 2.1.3 will lead to different system errors. To determine whether to select or reject a landmark point, we use the following method to evaluate the system error.

From congruent triangle rules (side-side-side), we hypothesize that if the triangles formed by three identical pairs of points selected from 2 images are congruent, the error in these three pairs of points is the smallest and is recorded as zero.

Then, we define the triangle error $E_t$ that satisfies:

$$E_t(L_1, L_2, L_3) = \|L_1 L_2\| + \|L_1 L_3\| + \|L_2 L_3\| + t \qquad (2-7)$$

$\|L_1 L_2\|$ is the percentage of the length difference of the line segment $L_1 L_2$ in 2 images:

$$\|L_1 L_2\| = \frac{\||\overrightarrow{P_{L1} P_{L2}}| - |\overrightarrow{R_{L1} R_{L2}}|\|}{0.5(|\overrightarrow{P_{L1} P_{L2}}| + |\overrightarrow{R_{L1} R_{L2}}|)} \qquad (2-8)$$

t reflects the similarity of 2 triangles:

$$t = (\text{Max}(\|L_1 L_2\|, \|L_1 L_3\|, \|L_2 L_3\|) - \text{Min}(\|L_1 L_2\|, \|L_1 L_3\|, \|L_2 L_3\|)) \qquad (2-9)$$

The above formulas indicate that, when the three edges of the corresponding triangles are equal in length in the two coordinate systems, $E_t = 0$. Conversely, when the lengths of the three edges are unequal, $E_t > 0$. The greater the length difference is, the greater $E_t$. Considering the parameter t, $E_t$ is smaller when the edges vary proportionally.

Then, the error of each landmark is judged. The error $E_i$ of a point i satisfies:

$$E_i = \sum_{u,v \in M} E_t(L_i, L_u, L_v) \qquad (2-10)$$

In this formula: M = {1,2,. . .,m }; m is the sum of pairs of landmarks; i,u,v∈M; u≠v≠i.

In the error evaluation system, the pairs of points with large errors will affect the pairs with small errors. However, the maximum error of points is always $E_{max} = \text{max}(E_i)$. Step-by-step sorting of $E_{max}$ can provide the error value of each pair of points. Finally, the three pairs of points with the least error are obtained and used as points $L_1, L_2$, and $L_3$ for the geometric transformation in the TriP method.

When given ≥4 pairs of landmarks, our program will choose the 3 best pairs with the least error to build the coordinate system N and will calculate system error with the 4th pair. The details of the algorithm are in Appendix 1.

## Analysis of factors affecting the system error of the TriP method

The effects of the number of landmarks provided and the distance between the selected points on system errors were analyzed.

The first factor was analyzed experimentally. We selected 4–16 pairs of points in turn for system error calculations and drew the variation curve for the average system error. Then, the paired-sample t-test was performed for the system errors of two adjacent groups with different numbers of landmarks to determine whether the system error differed. Subsequently, the distance between the selected points was analyzed with geometric calculation methods.

## Accuracy indicators: Selection and calculation

The first indicator selected in this study was the height of the cover screw, which helps to integrate different implants into the same system. The calculation and the method of correcting the height of the cover screw are provided in Appendix 1.

One study [2] indicated that the parameters describing position deviations between the planned implant (blue) and the real implant (red) are AD, CGD, coronal vertical deviation (CVD), coronal lateral deviation (CLD), AGD, apical vertical deviation (AVD), and apical lateral deviation (ALD) (Fig 1G).

Because of the underestimation of new bone by CBCT and the incomplete bone calcification, there is a discrepancy between the height of the bone surface from the CBCT data and the actual findings during the operation. During the manual insertion stage, the surgeon often adjusts the implant depth subjectively by observation, causing implant deviation via a nontemplate cause. Therefore, we proposed new indicators—coronal and apical rectified deviations (CRD and ARD, respectively)—to provide corrections for the subjective depth adjustments performed during manual implant insertion to simply analyze the accuracy of the implant template (Fig 1H). The details of the algorithm are in Appendix 1.

## Traditional image registration method

Traditional image registration was performed with Mimics software, version 10.01 (Materialise n. v., Belgium), and the same indicators were calculated. The detailed steps are shown in Appendix 1.

## Application and accuracy analysis of the Visual Direction-INdicating Guide (VDING)

This study aimed to verify the precision of the VDING template. The guide template is a depth-control template combined with a visual direction-indicating device (Fig 2A). This template was developed to accommodate the low angle restriction of a short guide sleeve when the interarch distance in the implantation area is slightly insufficient for a normal height template [8]. The design idea and fabrication method are shown in Appendix 1.

After the ethics examination and approval, we applied VDING. After obtaining informed consent, 13 patients with 15 missing teeth who planned to receive implantation at the Stomatology Center of Xiangya Hospital participated in the clinical study of the VDING template (cases and inclusion criteria appear in Appendix 1).

After jaw data collection with a CBCT machine (KaVo Dental GmbH, Germany; 0.25 mm slice interval), VDING templates were fabricated and tried in patients before operation. Before surgery, the guide template was sterilized under the conditions of normal temperature and atmospheric pressure by plasma sterilization (Sterrad 100S, Johnson and Johnson, USA). The operation was performed according to the guidelines for implant surgery (Fig 2B).

The patients in the study underwent CBCT examinations immediately after the operations in the examination conditions.

## Comparison and statistical analysis of the TriP and traditional methods

The individuals who analyzed the accuracy of the TriP method received training in selecting image registration landmarks. The accuracy analysts also passed a consistency evaluation before selecting landmarks. The landmark selection process was performed by one researcher with supervision by another researcher.

Sixteen pairs of landmarks were selected. The coordinates of the coronal endpoint $R_{IC}$ and the apical endpoint $R_{IA}$ of the real implant were recorded. The coordinates of the planned

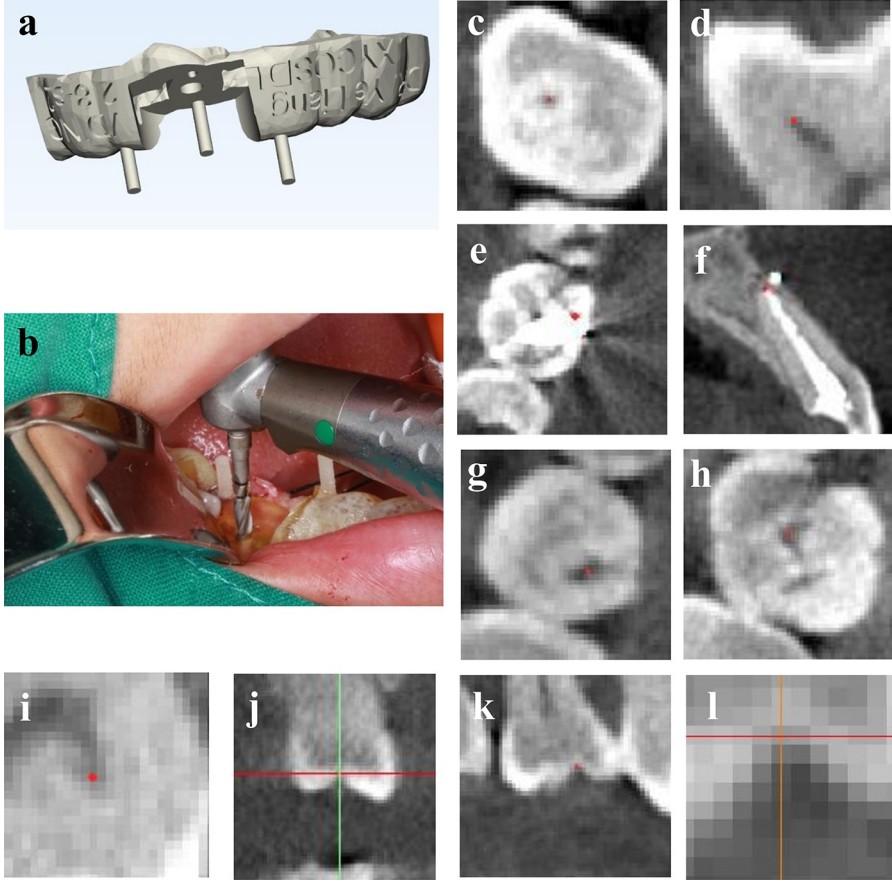

**Fig 2. VDING template and visuals used to select landmarks.** (A) Design of the VDING template. (B) The VDING template is used in implant surgery. The implant hand piece is controlled to make the drill needle parallel to the visual direction-indicating device. (C-L) Visuals for selecting landmarks in a CBCT image. (C) Enlarged dot-like pit in crown of posterior teeth. (D) Acute pulp angle. (E) Enlarged teeth after crown filling. (F) Teeth after root canal therapy. (G) Enlarged crown fissure. (H) Point on the triangular crack. (I) The pulp angle with an irregular shape. (J) Unenlarged pit. (K) The pit selected as a landmark. (L) After enlargement of the selected landmark in k, the vertex of the pit is still not clear enough.

implant endpoints $P_{IC}$ and $P_{IA}$ were output from the design software. The coordinates of these points were recorded and saved. Then, a program (https://github.com/coolleafly/VDING) written with the Python 3.6 platform was used for the calculations. The program outputs all the implantation accuracy indicators described in 2.1 and the system errors for each implant.

The traditional method was executed by the steps described in 2.2. The mean value of the measured implant position determined by two trained inspectors was taken.

To compare the two accuracy analysis methods, a normality test and a paired-sample t-test were performed to compare the implantation accuracy results by SPSS 21.0 statistical analysis software. The test level was set at $\alpha = 0.05$.

# Results

## Analysis of factors affecting system error

**Influence of number of provided landmark points on system error.** In the same cases, 4 to 16 landmarks were provided, and the system errors of the TriP image registration method with different numbers of landmarks are shown in Fig 3A.

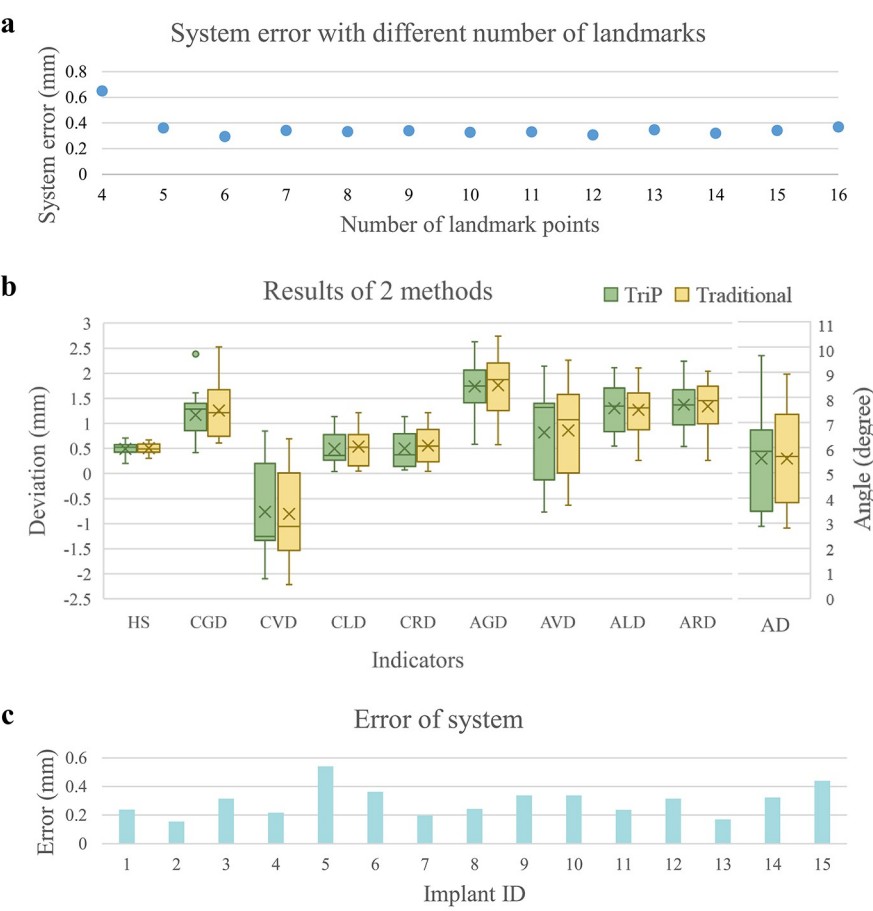

**Fig 3. Results.** (A) Influence of the number of provided landmarks on the system error. (B) The results for each indicator with the TriP method and the traditional method. (C) System error for each implant case with the TriP method.

A paired-sample t-test was performed to evaluate the system errors with two adjacent number of points. The results are as follows (Table 1):

**Influence of distance between the 3 selected pairs of landmark points on system error.** A geometric method was used to deduce that because the error in point selection is small and similar, increasing the distance between landmarks decreases the angle error in the geometric transformation (Fig 1F). The specific deduction process is shown in Appendix 1. The results indicate that selecting dispersed landmarks helps reduce the system error of the TriP method.

## Comparison of implantation accuracy calculation by the two methods

**Normality test.** The Shapiro-Wilk test was used to assess the normality of each implant accuracy indicator. All the indicators used in the two methods were normally distributed (P>0.05). Therefore, the mean value±standard deviation is used to describe each indicator.

**Comparison of the results for cover screw heights.** Using the calculation methods above, the results for each indicator with the TriP method and the traditional method were obtained and are shown in Fig 3B. The results of paired-sample t-tests comparing the 2 methods are shown in Table 2. For each indicator, the two methods showed no significant difference (P>0.05). The two image registration methods are considered consistent.

**Table 1. Paired-sample T-test results for the influence of provided number of landmarks on system error.**

| Pair | N | Mean±SD of the 1st group | Min—max of the 1st group | Mean±SD of the 2nd group | Min—max of the 2nd group | P |
|---|---|---|---|---|---|---|
| 4 landmarks—5 landmarks | 15 | 0.65±0.48 | 0.48–0.14 | 0.36±0.17 | 0.17–0.15 | 0.02* |
| 5 landmarks—6 landmarks | 15 | 0.36±0.17 | 0.17–0.15 | 0.30±0.10 | 0.10–0.16 | 0.29 |
| 6 landmarks—7 landmarks | 15 | 0.30±0.10 | 0.10–0.16 | 0.34±0.11 | 0.11–0.16 | 0.05 |
| 7 landmarks—8 landmarks | 15 | 0.34±0.11 | 0.11–0.16 | 0.33±0.11 | 0.11–0.16 | 0.24 |
| 8 landmarks—9 landmarks | 15 | 0.33±0.11 | 0.11–0.16 | 0.34±0.14 | 0.14–0.16 | 0.83 |
| 9 landmarks—10 landmarks | 15 | 0.34±0.14 | 0.14–0.16 | 0.33±0.11 | 0.11–0.16 | 0.66 |
| 10 landmarks—11 landmarks | 15 | 0.33±0.11 | 0.11–0.16 | 0.33±0.18 | 0.18–0.13 | 0.92 |
| 11 landmarks—12 landmarks | 15 | 0.33±0.18 | 0.18–0.13 | 0.31±0.18 | 0.18–0.11 | 0.37 |
| 12 landmarks—13 landmarks | 15 | 0.31±0.18 | 0.18–0.11 | 0.35±0.21 | 0.21–0.11 | 0.20 |
| 13 landmarks—14 landmarks | 15 | 0.35±0.21 | 0.21–0.11 | 0.32±0.19 | 0.19–0.12 | 0.33 |
| 14 landmarks—15 landmarks | 15 | 0.32±0.19 | 0.19–0.12 | 0.34±0.20 | 0.20–0.10 | 0.51 |
| 15 landmarks—16 landmarks | 15 | 0.34±0.20 | 0.20–0.10 | 0.37±0.23 | 0.23–0.13 | 0.55 |

*Significant difference between the two groups in the paired-sample t-test ($P < 0.05$).

**System error of the TriP method.** The image registration error of each implant was evaluated by the method in 2.1.5, as shown in Fig 3C. The overall system error for all the samples was 0.30±0.10 mm.

## Discussion

### System error and its optimization in accuracy calculation

Operator proficiency affects the accuracy of image registration, and the accuracy is not easy to judge. Without a system error value provided by analysis software, comparisons of accuracy between different studies can be questioned, and the registration process is hard to optimize.

To evaluate the registration system error objectively, the system error was calculated in our study. This calculation not only increases the reliability of calculating implant accuracy but also facilitates comparisons of implant accuracy statistics between different studies. Moreover, this evaluation allows researchers to identify landmarks that have low error and improve registration efficiency.

### Ways to reduce the system error in image registration

As indicated in 3.1.1, increasing the number of selected points does not improve the results. As the number of selected points increases, the system error approaches convergence. Selecting too many points causes problems: 1. The number of clear and sharp landmarks is insufficient; 2. Useless work increases; and 3. The distance between landmarks becomes too small. Our results indicate that the number of recommended landmarks is 5 to 7. According to the deduction in 3.1.2, a sufficient distance between landmarks is conducive to reducing system error.

Additionally, after many training processes, we recommend selecting the following landmarks to improve accuracy: 1. the enlarged dot-like pit in the crown of posterior teeth; 2. the spiculate pulp angle; and 3. the sharp tip of tooth roots (Fig 2C and 2D). After comparison, the following locations were eliminated: 1. a crown with a filling or restoration or a root with root canal treatment; 2. a fissure or triangular crack in the enlarged view; and 3. a pulp angle with an irregular shape (Fig 2E–2I). To improve the visualization of boundary lines, brightness and contrast can be changed, and a scaling function should be used (Fig 2J).

**Table 2. Paired-sample T-test for the results of the 2 methods.**

| Indicator | Image registration method | Mean (SD) | Min | Max |
|---|---|---|---|---|
| HS | TriP | 0.49±0.14 | 0.2 | 0.71 |
| HS | traditional | 0.5±0.1 | 0.3 | 0.67 |
| AD | TriP | 5.57±1.91 | 2.87 | 9.66 |
| AD | traditional | 5.57±1.81 | 2.81 | 8.92 |
| CGD | TriP | 1.17±0.50 | 0.42 | 2.38 |
| CGD | traditional | 1.26±0.55 | 0.61 | 2.53 |
| CVD | TriP | -0.76±0.85 | -2.1 | 0.85 |
| CVD | traditional | -0.81±0.92 | -2.22 | 0.69 |
| CLD | TriP | 0.49±0.33 | 0.04 | 1.13 |
| CLD | traditional | 0.54±0.38 | 0.05 | 1.21 |
| CRD | TriP | 0.50±0.35 | 0.07 | 1.14 |
| CRD | traditional | 0.56±0.38 | 0.05 | 1.21 |
| AGD | TriP | 1.74±0.51 | 0.58 | 2.63 |
| AGD | traditional | 1.76±0.55 | 0.58 | 2.74 |
| AVD | TriP | 0.82±0.85 | -0.77 | 2.14 |
| AVD | traditional | 0.86±0.91 | -0.64 | 2.26 |
| ALD | TriP | 1.30±0.50 | 0.55 | 2.11 |
| ALD | traditional | 1.27±0.52 | 0.26 | 2.11 |
| ARD | TriP | 1.37±0.49 | 0.54 | 2.24 |
| ARD | traditional | 1.34±0.52 | 0.26 | 2.04 |

## Possible sources of errors in the guided implantation process

Although the implant guide template provides a method to improve precision, the follow errors remain:

1. The following data collection errors remain that affect the guide template design. The limited spatial resolution of CBCT cause errors in selecting landmarks (Fig 2K and 2L). Slight patient motion leads to bias in CBCT sampling. CT images reflect new bone surfaces with bias [9]. Materials shrink during the plaster casting performed to take impressions [10].

2. When producing the guide template, the accuracy of 3D printing is limited, and the subsequent process produces further errors [11].

3. The bur-cylinder gap between the implant drill needle and the guide sleeve may cause mechanical errors during guided implantation [12].

4. Because of inhomogeneities in bone density, the implant tends to squeeze the low-density bone wall, leading to deviations. Manual implant insertion may lead to deviations [13].

5. Errors can be introduced during measurement. Existing cover screws, healing abutments and image registration can all lead to errors.

## Implantation accuracy analysis indicator and its function

Different analysis indicators have been proposed in different studies. In this study, the indicators AD, CGD, CVD, CLD, CRD, AGD, AVD, ALD, and ARD are considered to reflect the overall implant accuracy.

AD reflects the axial deviation and is the core indicator when analyzing the precision of the guide template. This indicator is related to the accuracy of the guide template position, the

length of the guide sleeve, the tightness of the guide sleeve, and the existence of other auxiliary directional devices.

CGD, CVD, CLD and CRD reflect the deviation during implant entry. CGD is the most intuitive and recognized overall position deviation indicator. When CLD and CRD are both large, the implant entry deviation is too large, most likely due to poor placement of guide templates. If CVD is large, and CRD is small, the unsatisfactory development of new bone on CBCT likely results in inaccurate implant depth in the surgical plan. If AD is small in this case, the guide template may still have good accuracy.

The indicators reflecting apical deviation are AGD, AVD, ALD and ARD. If AD and AGD are both small, the guide template is excellent. If there is some deviation but CRD and ARD are both small, the implant still moves along the predetermined axis and entry path, but the depth of the manual implant insertion differs from the planned depth. In this case, the guide template is still excellent and generally does not affect the important surrounding anatomical structures. AVD reflects the implant apical vertical deviation, which can result in injury of the inferior alveolar nerve or maxillary sinus. Therefore, the use of a guide template with a large AVD should maintain a sufficient safe distance. In contrast, if AVD is small, the safe distance can be reduced appropriately to decrease the operation complexity. ALD reflects the horizontal apical deviation, which can lead to damage to adjacent teeth and accidental penetration of cortical bone. Therefore, if ALD is large, the guide should be used cautiously in a narrow implantation area.

## Supporting information

**S1 File. Registration of the planned image and the real implant image.** Error analysis and calculation in the TriP image registration method Calculation and correction of cover Screw Height (HS) and other indicators Calculation of accuracy indicators CRD and ARD Traditional image registration method Design and fabrication of VDING guide template Accuracy evaluation for the VDING template Deduction of influence of the distance on system error Patient inclusion criteria and information.
(DOCX)

## Acknowledgments

We thank Professor ShengHui Liao for providing E-3D software and targeted optimization of the software. Thanks Professor JiaDa Li for revising the calculation of data and revising the manuscript critically for important intellectual content.

We are grateful for resources from the High Performance Computing Center of Central South University. We are grateful to American Journal Experts for English language editing, which contributed to the clarity of the manuscript.

## Author Contributions

**Conceptualization:** Ye Liang, ShanShan Yuan, JingJing Huan, ChangYun Fang, Jia-Da Li.

**Data curation:** Ye Liang, ShanShan Yuan, JingJing Huan, HuiXin Wang, YiYi Zhang, ChangYun Fang, Jia-Da Li.

**Funding acquisition:** Ye Liang, ShanShan Yuan.

**Investigation:** JingJing Huan.

**Methodology:** Ye Liang, ShanShan Yuan, ChangYun Fang.

**Resources:** Ye Liang, ShanShan Yuan, JingJing Huan, HuiXin Wang, YiYi Zhang.

**Software:** Ye Liang, ShanShan Yuan, Jia-Da Li.

**Supervision:** Ye Liang, JingJing Huan, ChangYun Fang, Jia-Da Li.

**Validation:** Ye Liang, ShanShan Yuan, JingJing Huan, HuiXin Wang, YiYi Zhang, ChangYun Fang, Jia-Da Li.

**Writing – original draft:** Ye Liang, ShanShan Yuan.

**Writing – review & editing:** Ye Liang, ShanShan Yuan, JingJing Huan, HuiXin Wang, YiYi Zhang, ChangYun Fang, Jia-Da Li.

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
