## [Decision Letter · Decision Letter 0]

14 Nov 2019

Use of Three Points to Determine the Accuracy of Guided Implantation

PONE-D-19-16430

Dear Dr. Fang,

We are pleased to inform you that your manuscript has been judged scientifically suitable for publication and will be formally accepted for publication once it complies with all outstanding technical requirements.

With kind regards,

Sompop Bencharit, DDS, MS, PhD, FACP

Academic Editor

PLOS ONE

1) Thank you for including your ethics statement:

"Human participants.

Ethics Commitee of Xiangya Hospital of CentrE South University

approval number: 201512515"

a. Please amend your current ethics statement to confirm that your named institutional review board or ethics committee specifically approved this study.

Reviewers' comments:

Reviewer's Responses to Questions

**Comments to the Author**

1. Is the manuscript technically sound, and do the data support the conclusions?

Reviewer #1: Yes

Reviewer #2: Yes

2. Has the statistical analysis been performed appropriately and rigorously? 

Reviewer #1: Yes

Reviewer #2: Yes

3. Have the authors made all data underlying the findings in their manuscript fully available?

Reviewer #1: Yes

Reviewer #2: Yes

4. Is the manuscript presented in an intelligible fashion and written in standard English?

Reviewer #1: No

Reviewer #2: Yes

5. Review Comments to the Author

Reviewer #1: The study was aimed at validating a method for determination of the accuracy of guided dental implantation. The topic is interesting and the methodology is appropriate. English grammar and wording may need revision.

Reviewer #2: This is an area that really needs to be standardized. We need a consistent way to evaluate our outcomes so that we can be sure new technology is delivering better patient results. An open source tool that can be used across multiple platforms would be ideal.

6. PLOS authors have the option to publish the peer review history of their article (what does this mean?). If published, this will include your full peer review and any attached files.

Reviewer #1: No

Reviewer #2: Yes: Daniel Whitley

---

## [Editor Report · Acceptance letter]

20 Nov 2019

PONE-D-19-16430 

Use of Three Points to Determine the Accuracy of Guided Implantation 

Dear Dr. Fang:

I am pleased to inform you that your manuscript has been deemed suitable for publication in PLOS ONE. Congratulations! Your manuscript is now with our production department. 

With kind regards,

on behalf of

Dr. Sompop Bencharit 

Academic Editor

PLOS ONE